# Biomechanical Comparison between Down-the-Line and Cross-Court Topspin Backhand in Competitive Table Tennis

**DOI:** 10.3390/ijerph19095146

**Published:** 2022-04-23

**Authors:** Kaige Xing, Lanping Hang, Zijun Lu, Chuangui Mao, Dong Kang, Chen Yang, Yuliang Sun

**Affiliations:** 1Department of Physical Education, Chang’an University, Xi’an 710064, China; kaigex@chd.edu.cn (K.X.); hlphlp@chd.edu.cn (L.H.); kangdong@chd.edu.cn (D.K.); 2Department of Exercise Science, School of Physical Education, Shaanxi Normal University, Xi’an 710119, China; luzijun@snnu.edu.cn (Z.L.); maocg117@snnu.edu.cn (C.M.); 3Department of Kinesiology and Physical Education, Faculty of Education, McGill University, Montreal, QC H2W 1S4, Canada

**Keywords:** table tennis, backhand, topspin, kinematics and kinetics, down-the-line and cross-court

## Abstract

The aim of this study was to compare the kinematic and kinetic differences of the racket arm when balls were hit cross-court (CC) and down the line (DL) by topspin backhand. Eight elite female players participated and were instructed to hit the ball down the line and cross-court using a topspin backhand. Kinematic and kinetic data were collected. The results show that at the impact, participants had a greater wrist flexion angle in DL than CC (*p* = 0.017). The angular velocity of shoulder flexion (*p* = 0.038), shoulder abduction (*p* = 0.006) and thorax–pelvis internal rotation (*p* = 0.017) was faster when participants impacted the ball DL than CC. As for the joint kinetics, the shoulder external rotation moment was greater in CC than DL (*p* = 0.043). For a high-quality DL technique, it is important to exhibit a greater wrist flexion and have faster adduction and flexion in the shoulder, as well as faster internal rotation in thorax–pelvis, while having a smaller wrist flexion and more external rotation power in the shoulder are important to perform a CC at the impact. If these key and different factors of hitting CC and DL are ignored, it may lead to failure to complete a high-quality shot.

## 1. Introduction

The technical skill of table tennis is one of the most crucial factors that determine the level of players, especially in high-level performance [1,2]. Topspin backhand in modern table tennis is commonly used in aggressive and offensive shots [3]. One previous study found that the backhand was a crucial technique for scoring by analyzing the strategy of Chinese professional athletes used in games, and more topspin backhands were used in female athletes [4]. The tactic of changing the line is usually used to gain scores in most women’s games, such as impacting the ball with a backhand against the ball to the opponent’s forehand. As the sudden change in the path of the ball may force the opponent to alter her movements, the opponent is more likely to mishit. Assessment of typical differences in the details of the technique may allow for optimizing and preparing appropriate training plans [5,6]. An in-depth understanding of topspin backhand kinematic and kinetic differences between the cross-court and down-the-line could allow players to impact the ball effectively to the different routes of the ping-pong table and may also be useful for players to predict the direction of the ball in advance when they do not know it, according to the specialized characteristics [7]. However, to our best knowledge, there is no research on the difference in topspin backhand between down-the-line (DL) and cross-court (CC) both in kinematics and kinetics.

There are various studies investigating the kinematics and kinetics of the backhand, most of which focused on the racket arm. Racket, joint and ball kinematics were compared between the shake-hand and pen-hold grips in backhand strokes. Greater ball and racket velocities were observed for the shake-hand grip [8]. The shake-hand grip also generally demonstrated decreased final trunk left rotation angles, decreased final shoulder abduction angles, increased trunk right rotation angular velocities, increased shoulder adduction angular velocities and increased forearm supination angular velocities [8]. The effect of the racket mass and the rate of strokes on kinematics and kinetics was analyzed in backhand strokes [9]. It was found that the racket speed at impact was significantly lower for high ball frequency than for low ball frequency. This was probably because pelvis and upper trunk axial rotations tended to be more restricted for the high ball frequency. The topspin and backspin were also compared by Lino [10]. They found that the racket upward velocity of a backspin backhand at impact was dramatically faster than that of a topspin backhand. The contribution of wrist dorsiflexion to upward velocity is greater in a backspin. In terms of angular velocity, elbow extension and wrist dorsiflexion at impact were not significantly different between the two types of backhands. The contribution of elbow extension and wrist dorsiflexion to the upward velocity is related to the angle, not to their angular velocity. Furthermore, the velocities of arm abduction and shoulder girdle rotation toward the playing side had a principal effect on racket velocity. The angular velocities of arm internal rotation and shoulder adduction are important components of coordinated stroke [11].

Some studies also focused on the generation and transmission of mechanical energy of the racket arm at the impact. Lino compared the kinetic differences between intermediate and advanced athletes in topspin forehand stroke [12]. The larger internal rotation torque in the shoulder was observed in advanced players. Because of the larger shoulder internal rotation torque, the mechanical energy from the trunk to the arm was transferred at a higher rate in advanced players than in intermediate players. The authors considered that it was an important factor that leads to the higher racket speed. They also found that energy transfer by the shoulder joint force in the vertical direction was the largest contributor to the mechanical energy of the racket arm in backhand strokes. The upward thrust of the shoulder and the late timing of the axial rotation of the upper trunk are important for an effective topspin backhand [13]. All these studies with the backhand mentioned above took the CC stroke into account or did not pay much attention to the target line of shooting. However, DL shots are also widely used in competitive table tennis [14].

Biomechanics studies not only have improved the performance of athletes, but they also helped players to have a better understanding of the technique. There were many comparative studies in table tennis in terms of kinematics and kinetics, such as comparisons between topspin and backspin, high performance and intermediate players, shake-hand and pen-hold grips, forehand and backhand, low-speed ball and high-speed ball, and female and male players. However, the subject that compares kinematics and kinetics differences between cross-court and down-the-line in topspin backhand has not yet been addressed in the existing literature. It is worth noting that differences between cross-court and down-the-line in kinematics have been reported in other sports and forehand strokes of table tennis. Lanzoni found that the angle of elbow flexion in DL was larger than that of CC at the moment of the maximum racket velocity, using forehand strokes [15]. Johannes Landlinger found that tennis players used the square stance significantly more often when playing the ball down the line and rotated their hips significantly further in the down-the-line situation [16]. These studies about analyzing differences between CC and DL in forehand can help others to understand this technique better.

Based on previous studies on the backhand and forehand in table tennis and the lack of knowledge of the key performance factors and expert discussions of topspin backhand hitting the ball to CC and DL, the aim of this study was to compare the kinematic and kinetic differences of the racket arm when balls were hit to the cross-court and down-the-line by topspin backhand. Due to impacting the ball to the different target positions of the table, it was hypothesized that there would be differences between cross-court and down-the-line in the values of angular parameters and angular velocity and therefore in the values of the moment.

## 2. Materials and Methods

Eight female advanced table tennis players (age: 19.8 ± 1.2 years, body mass: 63.0 ± 19.9 kg, body height: 166.7 ± 4.8 cm, training experience: 12.5 ± 2.5 years) participated in this study. They were all national level-I athletes in China. The sample size calculus was conducted in G*Power software (ver. 3.1.9.7; Heinrich-Heine-Universität Düsseldorf, Düsseldorf, Germany). One similar previous study has reported that the effect size was 1.37 for the elbow joint angle at the moment of impact [15]. When the power (1-β) was set to be 0.80 and the α level was set as 0.05, the sample size was at least 7 participants according to the calculation using G*Power. The subjects were members of the table tennis team at Shaanxi Normal University, Xian, China. They were all offensive players and used shake-hand grip rackets. During this experiment, no injuries occurred for at least one month before the test. All of the players agreed to voluntarily participate in the study and signed a written informed consent form. The experimental procedure was approved by a local ethics committee (No: 202016001 2020-09).

### 2.1. Procedure

The data collection took place in the biomechanical laboratory of Shanxi Normal University. A 10-camera motion analysis system (Oqus700+, Qualisys AB, Gothenburg, Sweden) was used to capture upper body kinematic data at a frequency of 200 Hz. The test set-up is shown in Figure 1. Since the distance between the body and the table was different when each subject played, the table tennis table was manually moved by an instructor according to each participant’s needs during the test.

During the data collection, athletes wore uniform tights and used the same racket (TIMO BOLL ZLC, Butterfly Technical Center, Tokyo, Japan) with the Butterfly Tenergy 64 (Butterfly Technical Center, Tokyo, Japan) and DHC Hurricane 3 (Double Happiness Sports Company, Shanghai, China) rubber sheets. To ensure the fairness of the test, the ball was served by a serving machine (Tai De V-989E, Tai De, Zhongshan, China). The ball speed, landing point, arc and frequency of the ball machine were set to the state closest to the game, and each tester received the same ball for testing. In order to obtain the ball speed and ball frequency similar to the game, we adjusted the speed and frequency of the serving machine to: ball speed upper wheel level 10 and lower wheel level 9 and ball frequency level 17. The width of the table tennis table was 1.525 m. There was a white line in the center of the table, called midline, which divides the two table areas into left (backhand) and right (forehand) parts, respectively. The landing point was in the backhand position, 45 cm from the midline.

After a warm-up, the participant was instructed to impact balls (D40+, Double Happiness Sports Company, Shanghai, China) to ensure that she was familiar with the experimental environment. Then, the subject stood in an anatomical posture to collect a static posture by the cameras. After that, the subject was asked to hit cross-court and down-the-line of topspin backhands (hitting two diagonal balls and two straight line balls) while the full-body kinematics were recorded. The data collection was finished when the player accurately impacted the left target (DL) 4 times and the right target (CC) 4 times. The instruction for this test was: “Imagine hitting the ball in the game”. Trials were recorded until the player correctly performed 10 shots for each situation. Three of the shots were analyzed for each condition of each subject.

A modified, full-body Helen Hayes marker system with marker clusters was used in addition to a custom-made model for the racket (57 reflective markers) (Figure 2). The kinematic data of the whole body were recorded, but this study only analyzed the motion of the racket limb and trunk.

The retroreflective markers were placed on the following body parts: left lower edge of scapula, right lower edge of scapula, pectoral cross, anterior sternum, left acromion, 3 marking points on the left upper arm, 2 on the lateral and medial of the elbow joint, 3 on the left forearm, 2 on the lateral and medial of the wrist joint, middle finger metacarpophalangeal joint of left hand. Right acromion, 3 on the right upper arm, 2 on the lateral and medial of the right elbow joint, 3 markers on the right forearm, 2 on the lateral and medial of the right wrist joint, middle finger metacarpophalangeal joint of right hand, left lateral anterior superior iliac spine, left posterior superior iliac spine, right posterior superior iliac spine, right anterior superior iliac spine. Five markers were on the racket, one was attached to the top of the racket, and the other four were attached to both sides of the racket.

### 2.2. Data Reduction

Preprocessed kinematics and kinetic data (C3D format) were imported to Visual 3D (V6.0, C-Motion, Germantown, MD, USA). The fourth-order Butterworth low-pass digital filter was used to filter the kinematic and kinetic data, and the cut-off frequency was set as 14 Hz and 100 Hz, respectively. Anatomical landmarks and segments were defined according to the Visual 3D framework model and the anthropometric data [17]. The inertial properties of the racket were also considered and calculated as a part of the hand.

Kinematic and kinetic data, including angle, angular velocity and joint moment for the shoulder, elbow, wrist and thorax–pelvis, were recorded. The wrist joint was defined as the angle between the forearm and the hand. The elbow joint was defined as the angle between the upper arm and the forearm. The shoulder joint was defined as the angle between the trunk and upper arm. The shoulder was considered as a ball-and-socket joint and had three degrees of motion, which were *X*-axis (the sagittal plane: flexion and extension), *Y*-axis (the coronal plane: adduction and abduction) and *Z*-axis (the transverse plane: internal rotation and external rotation). The wrist joint was considered as double-hinge joint and had two degrees of motion, which were *X*-axis (the sagittal plane: flexion and extension) and *Y*-axis (the coronal plane: adduction and abduction). The elbow was considered as double-hinge joint and had two degrees of motion, which were *X*-axis (the sagittal plane: flexion and extension) and *Z*-axis (the transverse plane: internal rotation and external rotation). Elbow rotation was regarded as forearm rotation in the current study. Thorax–pelvis was defined as the movement of the trunk relative to the pelvis, having motion of the flexion or extension, lateral flexion and rotation.

Biomechanical data such as joint angle and joint movement were exported by Visual 3D software. The data of angular velocity are calculated by formula: ∆θ and ∆t are the angle and time, respectively. ω = ∆θ/∆t denotes angular velocity. Moments were normalized by body weight and height (%BW·H). 

### 2.3. Movement Phase Definition

The backhand loop motion was divided into two stages: backward swing and forward swing. This paper analyzed lowest center of gravity of backward swing and the impact ball of forward swing. The swing stage refers to the time period between the lowest center of gravity and the impact. The value of swing phase was selected from the V3D and processed by the software named Origin 2017. The swing phase value of each participant was uniformly simulated into 100 values; then the average value of 8 participants was compared between DL and CC. Based on previous studies, the moment of maximum velocity of the racket was recognized as the moment of impact [15].

### 2.4. Statistics Analysis

For all variables, the mean values measured in the two examined conditions (CC and DL) were compared using paired *t* tests with SPSS statistics software (IBM SPSS Statistics 25, Armonk, NY, USA). The normality of distributions of differences was verified with Kolmogorov–Smirnov tests. Statistical significance was set at *p* = 0.05, and all the data are reported as mean ± standard deviation (SD). The effect size (ES) was calculated as Cohen’s d, where 0.2, 0.5, 0.8 and 1.2 represent small, medium, large and very large effect sizes, respectively.

## 3. Results

Significant differences were observed in elbow flexion, forearm internal rotation and shoulder abduction angles at the lowest gravity in Table 1. Elbow flexion angle was greater when playing DL than CC (*p* = 0.024, ES = 1.08). Furthermore, when playing to DL, shoulder abduction angle was greater than CC (*p* = 0.007, ES = 1.36). A greater forearm internal rotation angle was discovered in CC (*p* = 0.022, ES = 1.10). Wrist and thorax–pelvis angles had no significant differences at the lowest gravity.

Only the wrist joint had a significant difference at the impact, showing in Table 2. A greater wrist flexion was found in DL than CC (*p* = 0.017, ES = 1.17). Elbow, shoulder and thorax–pelvis had no significant difference at the impact.

In conclusion, compared to CC, elbow flexion and shoulder abduction angles were greater when balls were hit to DL during the backswing of the body’s lowest gravity, while the forearm internal rotation angle was greater in CC than DL. Only the sagittal plane of the wrist had a significant difference during the impact. Compared to CC, participants had a greater wrist flexion angle when they hit the ball to DL. The angles of the elbow, shoulder, thorax–pelvis and wrist adduction tended to be the same in the three axes at impact.

As shown in Table 3, no significant difference was found in the angular velocity at the lowest center of gravity, including wrist, elbow, shoulder and thorax–pelvis. By contrast, the difference in angular velocity was mainly reflected at the impact, as shown in Table 4.

Significant differences between the CC and DL were detected for the shoulder angular velocity in the coronal plane and the transverse plane at the impact. The angular velocity of shoulder flexion and shoulder adduction was greater in DL than in CC (*p* = 0.038, ES = 0.97; *p* = 0.006, ES = 1.38). The angular velocity of thorax–pelvis was left lateral flexion in CC, while the angular velocity of thorax–pelvis was right lateral flexion in DL (*p* = 0.019, ES = 1.13). In addition, the internal rotation of thorax–pelvis was faster in DL than CC (*p* = 0.017, ES = 1.15).

In conclusion, the angular velocity of the racket arm and thorax–pelvis tended to be similar at the lowest center of gravity when the ball was struck to the different lines, DL and CC. However, the angular velocity of DL and CC has respective characteristics at the impact. Compared to CC, the angular velocity of shoulder flexion, shoulder abduction and thorax–pelvis internal rotation was faster when participants impacted the ball to DL. Another feature of DL and CC was that the angular velocity was left lateral flexion in DL, while CC was right lateral flexion.

Although the moment of the wrist in CC was flexion and the moment in DL was extension in the sagittal plane, while the moment of the shoulder was external rotation in CC and the moment of the forearm was internal rotation in DL in the transverse plane, there was no statistical difference in the moment variable at the lowest center of gravity when the ball was hit to the DL and CC in Table 5.

At the impact, only the moment of the shoulder in the transverse plane had a significant difference. The shoulder external rotation moment was greater in CC than DL (*p* = 0.043, ES = 0.94), showing in Table 6.

In conclusion, there was no statistical difference in the moment variable when the balls were hit to the DL and CC at the lowest center of gravity. Furthermore, at the impact, only the moment of the shoulder external rotation had a significant difference.

The wrist was always maintained flexed and adduction angle both in DL and CC during the swing phase. As shown in Figure 3A, the flexion of the wrist reached the peak at 70% and then decreased gradually. A greater wrist flexed angle was observed in DL than LL during the whole swing phase. As shown in Figure 3B, the angle of wrist adduction was greater in CC during the whole process. Wrist adduction reached the peak at 20% in CC and reached the peak at 30% in DL; DL was prior to the peak. The adduction angle decreased down to the peak at 85%, then it increased immediately, and the adduction angle tended to increase when hitting the ball.

As shown in Figure 4A, the angle of elbow flexion hardly changed before 50% of the swing phase, while the flexion angle fell sharply after 50%. Even when the flexion angle decreased, the elbow was still in flexion. As shown in Figure 4B, the forearm was still the internal rotation; the angle of the forearm internal rotation was greater in CC than DL before 20% of the swing phase and then tended to be similar at the final impact.

The angle of the shoulder in the sagittal plane was flexion. Flexion had a steady rise at the beginning of the swing phase, as shown in Figure 5A. A greater shoulder abduction angle was observed in DL than CC during the whole swing phase, as shown in Figure 5B. The abduction angle in the shoulder was increased and reached the maximum at 80 % of the swing phase, and the abduction angle gradually decreased before the impact. The shoulder in the transverse plane was an internal rotation angle, which reached the peak before the impact, which was 70% of the swing phase, then decreased slowly, as shown in Figure 5C.

The thorax–pelvis angles in the sagittal plane, the coronal plane and the transverse plane were extension, right lateral flexion and internal rotation, respectively. The extension and right lateral flexion angle continually decreased from the beginning of the swing phase. Different from DL, the right lateral flexion in the CC decreased sharply at 70%, as shown in Figure 6B. The thorax–pelvis internal rotation was greater in CC than DL before 70% of the swing phase, but more internal rotation was observed in DL after 70%, as shown in Figure 6C.

The angular velocity of the wrist varied from flexion to extension and from adduction to abduction, which means that the angular velocity of the wrist does not always maintain one angular velocity both in the sagittal plane and the coronal plane during the whole swing stage. As shown in Figure 7A, the wrist angular velocity was flexion and increased before 50% of the swing phase; then it fell sharply and became the extension angular velocity at 65%. The angular velocity of wrist extension tended to slow down before the impact, but it was still the extension angular velocity. As shown in Figure 7B, the wrist angular velocity was adduction at the lowest center of gravity; then it changed to the abduction angular velocity and reached a maximum at 55% of the swing phase. Finally, it became adduction angular velocity at the impact.

The angular velocity of the forearm fluctuated sharply in the transverse plane during the swing phase, especially in DL, as shown in Figure 8B. The forearm was internal rotation angular velocity in DL during the whole swing phase, while the angular velocity varied from external rotation to internal rotation in CC. The extension angular velocity of the elbow was rapidly increased, as a faster extension angular velocity in CC was observed after 65%, as shown in Figure 8A.

As shown in Figure 9A, the shoulder was flexion angular velocity during the swing phase, and the flexion angular velocity was greater in DL than CC. As shown in Figure 9B, the shoulder angular velocity was adduction at the beginning and turned to the abduction before the impact. As illustrated in Figure 9C, the shoulder was internal rotation angular velocity in the transverse plane both in DL and CC; then internal rotation angular velocity decreased and turned to the external rotation at 65% of the swing phase before the impact.

As shown in Figure 10A, the thorax–pelvis was flexion angular velocity during the swing phase. The flexion angular velocity sharply decreased in DL at 70% of the swing phase, while the flexion angular velocity increased continually in CC until 80%. As illustrated in Figure 10B, a left lateral flexion angular velocity was observed in DL; however, the thorax–pelvis had left and right lateral flexion angular velocity in CC, and the left lateral flexion decreased sharply at 70% and then turned to right lateral flexion before the impact. As shown in Figure 10C, the thorax–pelvis angular velocity was internal rotation during the swing phase. The internal rotation angular velocity increased sharply in DL, while in CC, it increased slowly.

The wrist had the flexion and extension moment in the sagittal plane. As shown in Figure 11A, the DL was the extension moment and the CC was the flexion moment at the lowest moment of the center of gravity. Before the impact, the direction of the moment changed to flexion. The wrist had the adduction and abduction moment in the coronal plane, as shown in Figure 11B. The abduction moment was at the lowest center of gravity. Then, the abduction moment decreased and became the adduction moment with the forward swing. The adduction moment reached the peak at 70% in DL, and that of the CC reached the peak at 80% during the swing phase; DL peaked before CC. Before the impact, the direction of the moment changed to abduction, which was greater in CC than DL.

As shown in Figure 12A, the elbow was a flexion moment in the beginning then turned to the extension moment at 45%. Finally, the flexion moment of the elbow was in CC, while the extension moment was in DL. As illustrated in Figure 12B, the maximum external rotation of the forearm moment was greater in CC. The forearm external rotation moment increased at 65% of the swing phase then decreased and became an internal rotation moment.

The shoulder moment varied from flexion to extension in the sagittal plane and varied from the abduction to adduction moment in the coronal plane. The shoulder was always external rotation in the transverse plane during the whole swing phase, as shown in Figure 13C. The maximum external moment was greater in CC than DL. The external rotation moment decreased both in DL and CC before the impact, but it was still the external moment.

As shown in Figure 14A, the thorax–pelvis was always flexion moment. Although the thorax–pelvis was left lateral flexion moment, it had largely fluctuated both in the CC and DL, as shown in Figure 14B. As illustrated in Figure 14C, the thorax–pelvis had internal rotation and external rotation moments.

## 4. Discussion

The purpose of this study was to compare the joint angle, angular velocity and moment of the racket arm differences in backhand topspin strokes between the cross-court and down-the-line. We found that having a more flexed wrist angle at the impact may be better for those who want to hit the ball to the DL, while if players want to hit the ball to the CC, a less flexed wrist can more easily hit the ball to CC. The lowest center of gravity of the backswing is as important as the impact of the forward swing, greater elbow flexion and shoulder abduction angles are conducive to preparing to hit the ball to the DL, and the CC requires a greater forearm internal rotation angle. We also found that DL took less time than CC during the whole swing phase, and all the results that had a significant difference were faster in DL than CC. In the angular velocity, the most important result was thorax–pelvis. DL was left lateral flexion angular velocity, while the CC was the right lateral flexion, which has never been found in previous studies.

### 4.1. Joint Angle

Visual attention plays an important role in sports: athletes must observe the movement and position of athletes, as well as the position of the ball, in order to accurately predict the development of the game [7]. In sports, the angle of the body joints gives people the most intuitive feeling, and thus the player can predict the direction of the ball in advance according to the specialized body language [18]. An appropriate angle of the body can also allow an effective stroke, which may be the reason why most of the kinematic studies included the angle parameter.

In the current study, the angle of elbow flexion and shoulder abduction was greater in DL than CC at the lowest center of gravity of the backswing. It indicates that if players want to successfully hit the ball to DL, having a larger elbow flexion and shoulder abduction angle would be better during the backswing. In addition, the forearm internal rotation angle was greater in CC than DL at the lowest center of gravity, which means that more internal rotation in the forearm can better prepare the athlete to hit the ball to CC.

At the impact, the joint angle differences between CC and DL in the backhand topspin were only the wrist in the sagittal plane. Compared with CC, a greater wrist flexion angle was observed in DL. Wrist flexion may make it easier to hit the ball in a straight line. The racket’s face angle at the impact may be a key factor determining the trajectory of the ball [19]. The racket angle controlled by the wrist can form a specific angle between the ball and the racket in the process of hitting; that is, the racket will cover the ball and hit the right side of the ball so as to straighten the ball’s trajectory. On the contrary, taking right-handed athletes as an example, the cross-court target area is in the athlete’s right oblique front of the body; that is, when the wrist flexing is reduced, the trajectory of the racket face will also change correspondingly. This difference between the wrist in CC and DL could be due to the final direction of the shot. No significant difference was observed in the shoulder, elbow and thorax–pelvis at the impact. The lack of difference demonstrates that the players impact the ball with a similar shoulder, elbow and thorax–pelvis angle regardless of where the target position players want to hit is. Perhaps these factors are common and crucial in the coordinated swing phase of the backhand stroke movement.

Many research studies on various sports have found that there was little difference in kinematic parameters of elite athletes at critical moments such as impact. High-level table tennis players can play the ball with coordinated movements and can make the opponent in the game unable to predict the ball [20]. Therefore, the fact that only the wrist had a difference at the impact seems to be reasonable in the current study because there are fewer angle differences in the upper arm at the impact and the opponent may not know where the athlete wants to hit, increasing their chance of winning, which agrees with the previous study mentioned before.

As for the hip and trunk, Seemiller and Holowchak found that the topspin forehand required more rotation in the hip and trunk, whereas the topspin backhand needed less hip and trunk rotation [21]. The recent study supported that view, but it does not mean that trunk rotation is not important in the backhand. One should interpret this relatively low percentage with caution since trunk rotation is probably the most crucial factor in the development of racket speed [16]. Lino found that the rotation of the pelvis and upper trunk in the backhand may be limited under the condition of high ball velocity frequency [9]. However, in the current study, the serve frequency was adjusted to be close to the game, which may be the reason for the small rotation of the truck.

### 4.2. Angular Velocity

In the current study, more differences between DL and CC were found in angular velocity than the angle variable at the impact. The angular velocity reflected the different contributions of each link of the human body to the impacting quality. It is worth noting that the angular velocity of the backhand in the sagittal plane was higher than that in the coronal plane and transverse plane, which may be due to the motion trajectory of the backhand by the backward-to-forward swing. Some studies believe that the backhand required less left and right trunk movement and rotation, which may also be the reason for the faster angular velocity of the backhand in the sagittal plane [21]. Differences between DL and CC were mainly in the shoulder and thorax–pelvis, and there was no statistical difference in the wrist and elbow at the impact.

Previous investigations indicated that the shoulder flexion and external rotation, elbow extension and wrist dorsiflexion angle velocities were the main contributors to the racket forward velocity at impact [3,10]. Changing the type of topspin strike requires changing the time, speed, main distance parameters and the direction of the racket [22]. In the current study, it was observed that the angular velocity of the racket arm was different when the ball was hit to CC and DL. We found that for the DL mode, the angular velocities of the extension and adduction of the shoulder, as well as thorax–pelvis internal rotation and thorax–pelvis left lateral flexion, were faster than those of the CC mode. Interestingly, all these angular velocities that had differences were faster in DL than CC. Our findings also show that in the swing stage from the lowest center of gravity to the impact, DL took less time than CC. This trend leads to the assumption that a faster angular velocity is beneficial for pre-stretching the racket arm muscles. The swing phase took less time in DL; therefore athletes need to pull muscles rapidly in the limited time so as to acquire more elastic energy to effectively impact the ball. As previously stated by Elliott [23], a fluid transfer and a brief pause between the backswing and the forward swing were important because they make better use of the elastic energy stored (stretch–shorten cycle) and can increase the speed of segments [24].

In the current study, greater flexion and adduction angular velocities in the shoulder were found in the DL. As a strong fulcrum for elbow extension, the shoulder has more flexion, and the upper limb will be able to rotate more easily due to the reduced moment of inertia [21]. Similarly, a slow shoulder adduction not only affects the forward swing of the shoulder and elbow but also deflects the desired target position. When hitting the ball to DL, a quick shoulder adduction provides a strong guarantee of forward swing speed. In addition, in a study with a diagonal line as the target line, it was pointed out that in a backhand shot, the velocity of arm abduction and the rotation of the shoulder girdle toward the playing side, that is, the velocity of shoulder external rotation, is related to the racket velocity [11]. In this study, it was observed that the angular velocity of shoulder external rotation was more prominent in the CC, which was consistent with previous studies. However, it does not involve the index of racket speed, which needs follow-up research. The differences in the angular velocity of the racket arm can be explained as that DL is more like pushing the arm straight and quickly in the direction of the shot, such as the angular velocity of flexion, extension and adduction. The swing trajectory in CC is the right forward of the body, and thus the angular velocity of the racket arm was greater in the outward direction, such as abduction and external rotation.

Trunk rotation has been shown to play an important role in table tennis [8]. Some studies have shown that trunk movement is strongly correlated with racket velocity [25,26]. The most important and typical results of this present study in angular velocity are the internal rotation and lateral flexion thorax–pelvis. It is worth noting that the results show that the angular velocity of the thorax–pelvis was right lateral flexion in CC, while it was the left lateral flexion in DL, and the angular velocity of the thorax–pelvis internal rotation in DL was higher than CC. These changes may be a typical difference between CC and DL, which has not been found in previous studies.

Table tennis players use essentially the same standing position when playing topspin backhand regardless of CC or DL, which requires rapid adjustment of the body posture of upper limbs and trunk to achieve ideal hitting. The angular velocity combination of left lateral flexion and internal rotation in thorax–pelvis can make the body tilt slightly in front of the arm, placing it in a better position to play a straight line. The thorax–pelvis internal rotation angular velocity was slower in CC than DL; CC may need external rotation angular velocity in the follow-up swing which is why the thorax–pelvis internal rotation angular velocity was slow.

### 4.3. Moment

Table tennis is defined as a sport that needs fine muscle control due to the small field and the diversity of technology. High-level athletes can exhibit a finer degree of kinesthesia compared to non-experts [27]. The results show that only the moment of the shoulder joint in the transverse plane had a significant difference; the external rotation moment was greater in CC than DL. There is also no difference in the moment of the lowest center of gravity. Although there were fewer differences between DL and CC in moment, it showed some common rules, which can provide a reference for future teaching and practice.

The shoulder is essentially the end point of the trunk, and trunk rotation influences the forward movement of the shoulder in a positive way. In the current study, CC had more external rotation moment in the shoulder. This might be explained by that the diagonal line is on the right side of the body, and thus the shoulder external rotation can exert more moment.

The positive and negative changes were found in the wrist, elbow and shoulder during the swing phase, as shown in Figure 11, Figure 12 and Figure 13. Furthermore, most of the change range from positive to negative emerged before the impact, which may have been caused by the braking phenomenon in motion. The motion of the objects can be slowed or stopped by braking. For ball games such as tennis and table tennis, hitting is carried out when the ball rolls [28]. The braking can enable athletes to use the whole arm to hit the ball, rather than simply using the wrist or any joint, which can maintain the hitting stability [29]. Therefore, taking the wrist as an example, if the wrist moment was always extension in the sagittal plane, the wrist will over-stretch and lead to the wrong action such as the “shake wrist” that often occurs in beginners [30]. In order to prevent this situation, the abduction moment of the wrist gradually decreased before hitting the ball and became the flexion moment. The purpose of this is to keep the arm stable and shoot accurately.

In the practice of topspin backhand, beginners may perform the wrong movement. For instance, there is a common mistake that the shoulder and elbow are too close to the trunk when back-swinging the racket, which is caused by excessive shoulder extension moment and insufficient abduction moment. These wrong movements occur because there is no coordination between the moments in the racket arm. According to the backhand movement, the moment of the shoulder abduction and elbow flexion allows the arm to be in the most suitable position for a backward swing. The shoulder abduction moment gradually changed into the adduction moment before hitting the ball, as shown in Figure 13B. It was reasonable as if the shoulder continues to exert abduction moment when hitting the ball, it will shift the swing direction of the entire arm to the right (right-handed athletes, for example), which will have a negative effect. These results can provide a reference for future teaching and practice.

### 4.4. Limitation

One limitation of this study was that all the participants were female advanced players. It remains unknown whether the results obtained in this study are applicable to other players, such as male players. Another limitation was the small sample size. Only eight players participated in this study. With a larger sample size, the results would be more persuasive. The third limitation was that the participants hit the ball using the same racket, all participants had more than thirteen years of training experience, and we believe that they were able to hit the backhand their natural way. However, slightly different results might be gained using each play’s racket. Lastly, in order for the test to be replicated, it would be better to obtain the actual speed, arc and frequency of the ping-pong ball via some advanced instruments in the future.

## 5. Conclusions

The study may be beneficial for coaches and players to pay more attention to the wrist flexion angle in the DL and the shoulder external rotation moment in CC. In addition, in the angular velocity of the upper limb, the angular velocity of shoulder flexion and adduction was faster in DL, and the angular velocity of the internal thorax–pelvis was also faster in DL. It is also worth noting that DL was the left lateral flexion thorax–pelvis angular velocity, while CC was right lateral flexion. These technique differences between the two shooting targets could be used to help coaches and table tennis athletes optimize performance in both training and competition. What is more, from the perspective of future player development, it is absolutely necessary to integrate scientific research with the existing coaching manual.

## Figures and Tables

**Figure 1 ijerph-19-05146-f001:**
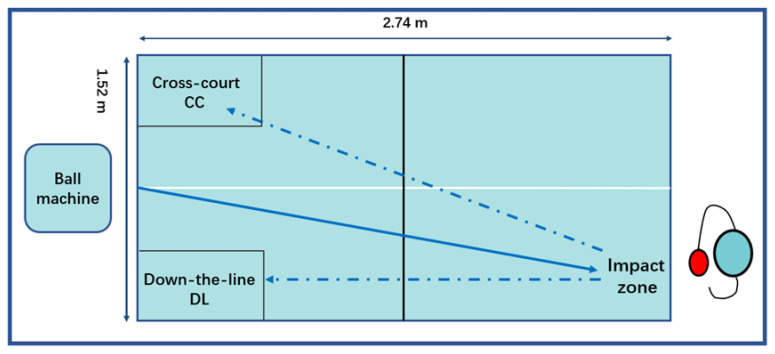
Experimental set-up of the study. CC = cross-court, DL = down-the-line. The solid line is the track of the ball projected by the machine, and the dash-dotted line is the track of the ball returned by the player.

**Figure 2 ijerph-19-05146-f002:**
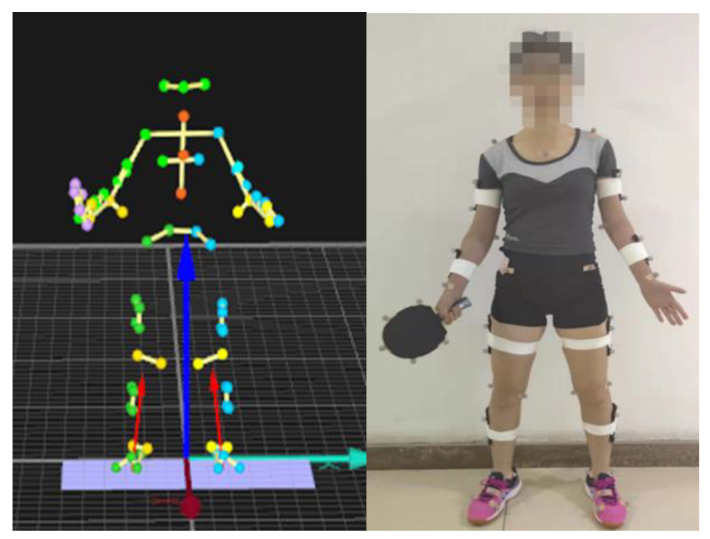
Marker placement adopted in the study.

**Figure 3 ijerph-19-05146-f003:**
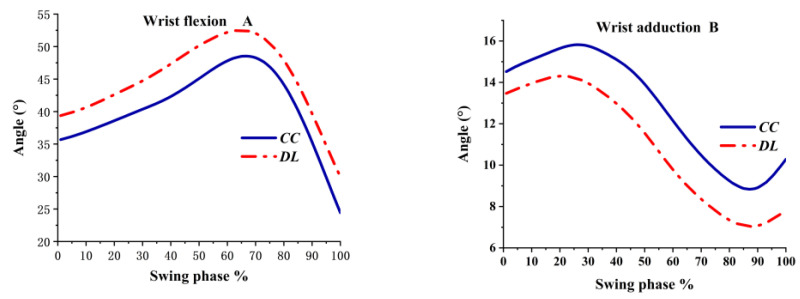
Wrist angles during the swing phase.

**Figure 4 ijerph-19-05146-f004:**
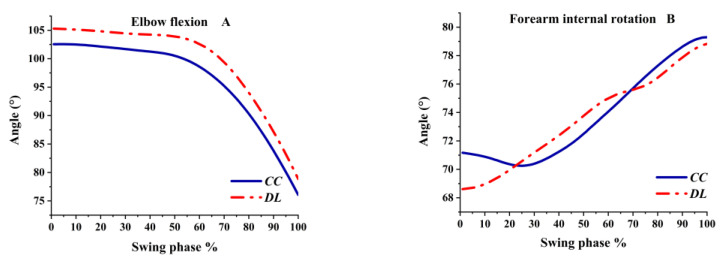
Elbow angles during the swing phase.

**Figure 5 ijerph-19-05146-f005:**
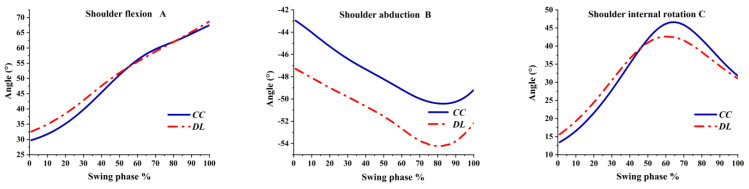
The shoulder angles during the swing phase.

**Figure 6 ijerph-19-05146-f006:**
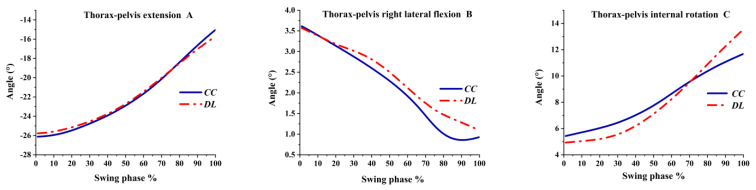
The pelvis–thorax angles during the swing phase.

**Figure 7 ijerph-19-05146-f007:**
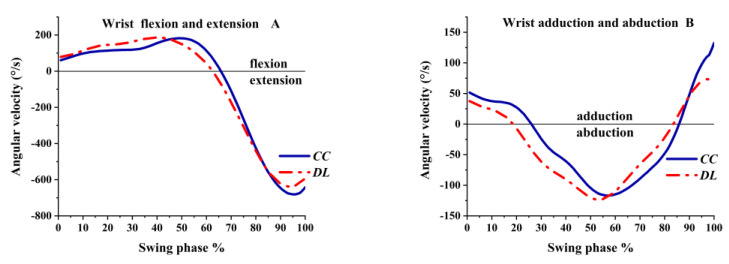
The angular velocity of the wrist during the swing phase.

**Figure 8 ijerph-19-05146-f008:**
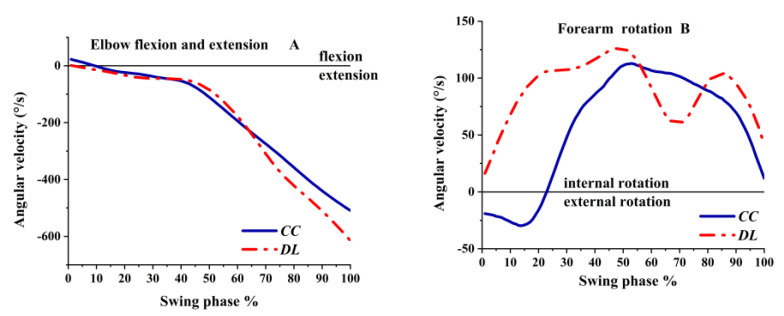
The angular velocity of the elbow during the swing phase.

**Figure 9 ijerph-19-05146-f009:**
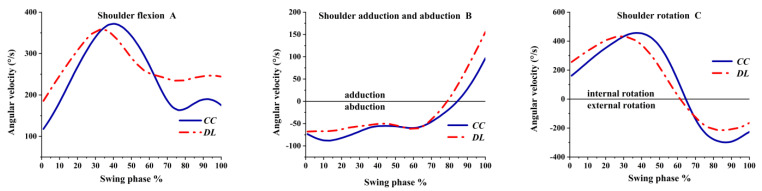
The angular velocity of the shoulder during the swing phase.

**Figure 10 ijerph-19-05146-f010:**
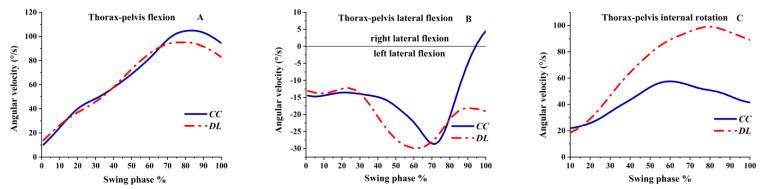
The angular velocity of the thorax–pelvis during the swing phase.

**Figure 11 ijerph-19-05146-f011:**
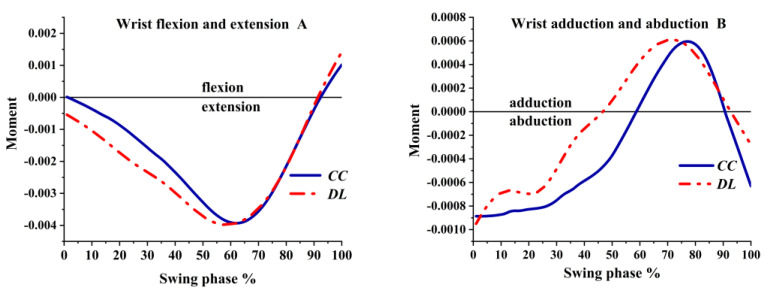
The moment of wrist joint during swing phase.

**Figure 12 ijerph-19-05146-f012:**
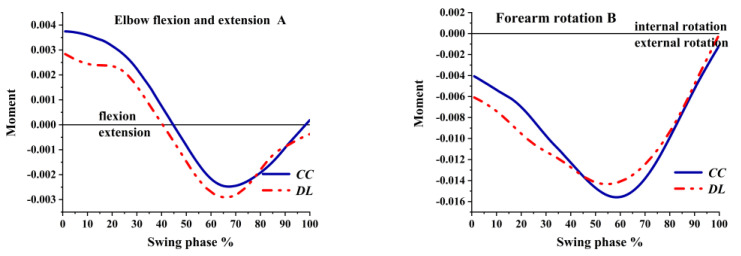
The moment of elbow joint during swing phase.

**Figure 13 ijerph-19-05146-f013:**
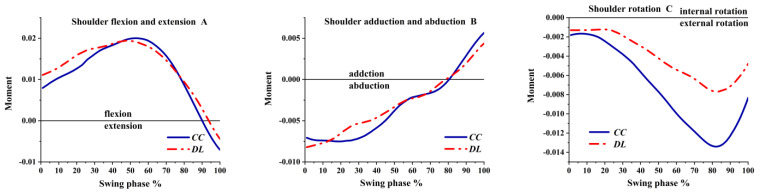
The moment of shoulder joint during swing phase.

**Figure 14 ijerph-19-05146-f014:**
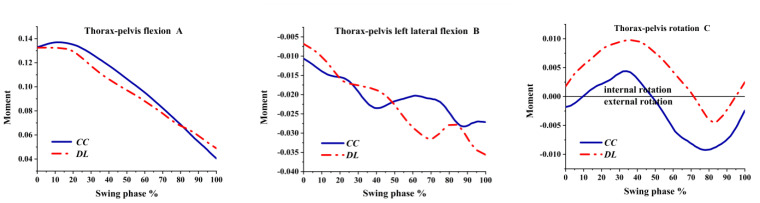
The moment of thorax–pelvis during swing phase.

**Table 1 ijerph-19-05146-t001:** The joint angle biomechanical variables of upper body at the moment of lowest center of gravity (*n* = 8).

Angle Variable	Cross-Court	Down-the-Line	*p*	ES
Wrist flexion (°)	35.69 ± 19.54	39.37 ± 18.62	0.069	0.82
Wrist adduction (°)	14.52 ± 14.00	13.47 ± 13.73	0.327	0.49
Elbow flexion (°)	102.55 ± 12.63	105.29 ± 12.61	0.024 *	1.08
Forearm internal rotation (°)	71.16 ± 15.24	68.61 ± 15.27	0.022 *	1.10
Shoulder flexion (°)	29.75 ± 12.79	32.56 ± 11.05	0.208	0.75
Shoulder abduction (°)	−42.96 ± 4.09	−47.28 ± 4.67	0.007 *	1.36
Shoulder internal rotation (°)	13.45 ± 15.92	15.61 ± 21.16	0.327	0.41
Thorax–pelvis extension (°)	−27.32 ± 3.31	−25.88 ± 5.51	0.804	0.43
Thorax–pelvis right lateral flexion (°)	3.61 ± 3.04	3.57 ± 2.57	0.938	0.03
Thorax–pelvis internal rotation (°)	5.11 ± 6.51	4.03 ± 2.72	0.484	0.18

* Indicates there was a significant difference in this variable between cross-court and down-the-line.

**Table 2 ijerph-19-05146-t002:** Joint angle biomechanical variables of upper body at the impact (*n* = 8).

Angle Variable	Cross-Court	Down-the-Line	*p*	ES
Wrist flexion (°)	23.37 ± 14.36	29.01 ± 11.05	0.017 *	1.17
Wrist adduction (°)	10.47 ± 4.66	7.89 ± 8.64	0.208	0.50
Elbow flexion (°)	75.25 ± 15.21	77.96 ± 12.88	0.401	0.42
Forearm internal rotation (°)	79.30 ± 15.88	78.87 ± 15.82	0.889	0.15
Shoulder flexion (°)	67.75 ± 13.53	69.11 ± 16.91	0.327	0.25
Shoulder abduction (°)	−49.03 ± 6.64	−51.94 ± 6.51	0.161	0.68
Shoulder internal rotation (°)	31.38 ± 15.39	30.75 ± 12.69	0.779	0.13
Thorax–pelvis extension (°)	−15.74 ± 6.03	−16.05 ± 5.79	0.327	0.20
Thorax–pelvis right lateral flexion (°)	0.75 ± 4.73	0.70 ± 3.38	0.674	0.03
Thorax–pelvis internal rotation (°)	11.01 ± 7.93	13.58 ± 8.54	0.069	1.09

* Indicates there was a significant difference in this variable between cross-court and down-the-line.

**Table 3 ijerph-19-05146-t003:** The angular velocity biomechanical variables of upper body at the moment of lowest center of gravity (*n* = 8).

Angular Velocity Variable	Cross-Court	Down-the-Line	*p*	ES
Wrist flexion (°/s)	61.01 ± 58.43	79.55 ± 50.26	0.282	0.36
Wrist adduction (°/s)	51.50 ± 74.03	37.50 ± 72.70	0.216	0.53
Elbow flexion (°/s)	22.55 ± 114.93	1.041 ± 146.13	0.575	0.29
Forearm external/internal rotation (°/s)	−19.05 ± 74.74	16.26 ± 101.73	0.546	0.25
Shoulder flexion (°/s)	117.93 ± 125.30	185.75 ± 213.04	0.372	0.19
Shoulder abduction (°/s)	−73.51 ± 86.86	−67.81 ± 71.83	0.633	0.19
Shoulder internal rotation (°/s)	161.85 ± 116.69	256.25 ± 196.88	0.261	0.48
Thorax–pelvis flexion (°/s)	10.27 ± 13.50	14.05 ± 18.10	0.639	0.19
Thorax–pelvis left lateral flexion (°/s)	−14.46 ± 20.42	−13.05 ± 18.38	0.830	0.08
Thorax–pelvis internal rotation (°/s)	21.47 ± 30.53	15.17 ± 34.67	0.318	0.42

**Table 4 ijerph-19-05146-t004:** The angular velocity biomechanical variables of upper body at the impact (*n* = 8).

Angular Velocity Variable	Cross-Court	Down-the-Line	*p*	ES
Wrist extension (°/s)	−625.59 ± 348.30	−576.64 ± 272.88	0.322	0.16
Wrist adduction (°/s)	143.17 ± 345.02	64.86 ± 185.63	0.575	0.42
Elbow extension (°/s)	−514.86 ± 157.86	−626.08 ± 186.90	0.386	0.89
Forearm internal rotation (°/s)	5.13 ± 259.11	33.40 ± 228.33	0.779	0.22
Shoulder flexion (°/s)	171.63 ± 94.82	243.59 ± 97.20	0.038 *	0.97
Shoulder adduction (°/s)	104.41 ± 81.94	163.97 ± 81.19	0.006 *	1.38
Shoulder external rotation (°/s)	−219.12 ± 159.17	−156.65 ± 162.73	0.239	0.48
Thorax–pelvis flexion (°/s)	93.43 ± 31.12	81.78 ± 35.98	0.386	0.36
Thorax–pelvis lateral flexion (°/s)	5.15 ± 30.73	−19.22 ± 40.33	0.019 *	1.13
Thorax–pelvis internal rotation (°/s)	41.16 ± 52.27	88.45 ± 67.49	0.017 *	1.15

* Indicates there was a significant difference in this variable between cross-court and down-the-line.

**Table 5 ijerph-19-05146-t005:** The joint moment of upper body at the moment of lowest center of gravity (*n* = 8).

Joint Moment	Cross-Court	Down-the-Line	*p*	ES
Wrist flexion and extension	0.0000094 ± 0.00103	−0.0005388 ± 0.00160	0.210	0.54
Wrist abduction	−0.00088 ± 0.00041	−0.00096 ± 0.00077	0.652	0.17
Elbow extension and flexion	−0.00378 ± 0.00159	0.00283 ± 0.00242	0.171	0.54
Forearm external rotation	−0.00400 ± 0.00376	−0.00610 ± 0.00494	0.248	0.17
Shoulder flexion	0.00794 ± 0.00642	0.01110 ± 0.00894	0.290	0.45
Shoulder abduction	−0.00706 ± 0.00356	−0.00820 ± 0.00349	0.385	0.36
Shoulder external and internal rotation	−0.00170 ± 0.00300	0.00134 ± 0.00413	0.744	1.17
Thorax–pelvis flexion	0.13291 ± 0.03063	0.13248 ± 0.02660	0.945	0.02
Thorax–pelvis left lateral flexion	−0.01065 ± 0.01980	−0.00682 ± 0.01369	0.267	0.47
Thorax–pelvis external and internal rotation	−0.00180 ± 0.01481	0.00178 ± 0.01471	0.252	0.49

**Table 6 ijerph-19-05146-t006:** The joint moment of upper body at the impact (*n* = 8).

Joint Moment	Cross-Court	Down-the-Line	*p*	ES
Wrist flexion	0.00109 ± 0.00104	0.00151 ± 0.00103	0.164	0.61
Wrist abduction	−0.00064 ± 0.00076	−0.00031 ± 0.00064	0.200	0.55
Elbow flexion and extension	0.00023 ± 0.00307	−0.00023 ± 0.00260	0.515	0.26
Forearm external/internal rotation	−0.00083 ± 0.00378	0.00024 ± 0.00307	0.094	0.75
Shoulder extension	−0.00759 ± 0.00654	−0.00503 ± 0.00356	0.314	0.43
Shoulder adduction	0.00587 ± 0.00384	0.00457 ± 0.00362	0.468	0.30
Shoulder external rotation	−0.00801 ± 0.01018	−0.00443 ± 0.00822	0.043 *	0.94
Thorax–pelvis flexion	0.04043 ± 0.01981	0.04887 ± 0.02441	0.160	0.61
Thorax–pelvis left lateral flexion	−0.02714 ± 0.01779	−0.03562 ± 0.02211	0.070	0.82
Thorax–pelvis external and internal rotation	−0.00240 ± 0.01972	0.00250 ± 0.01772	0.284	0.45

* Indicates there was a significant difference in this variable between cross-court and down-the-line.

## Data Availability

Not applicable.

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
