# Peer review of "Biomechanical Comparison between Down-the-Line and Cross-Court Topspin Backhand in Competitive Table Tennis"

_ijerph, 2022, doi:10.3390/ijerph19095146_

Round 1

Reviewer 1 Report

Dear Authors,

Thanks for giving me the chance to read this manuscript, “Biomechanical comparison between down-the-line and cross-court topspin backhand in competitive table tennis”. The current paper tries to compare the kinematic and kinetic of the racket arm differences when the balls were hit to the cross court and down-the-line using the topspin backhand.

Generally, it is a significant and timely topic in the field of physical training. However, there are some issues that should be appropriately addressed.

  1. Table presentation

  • In Tables 1, 2, and 3, it is unclear the unit for each variable.
  • Are Wrist flexion, Wrist adduction, Elbow flexion... Thorax-pelvis internal rotation without any units? If so, the authors should carefully explain the reasons and references to help readers have a better understanding of the current setting.

  1. Language and reference issue

  • The current manuscript needs intensive proofreading which could increase the readability of this paper.
  • Some references are missing in the current manuscript. For example, lines 49-50. The authors are advised to double-check the references in the current version.

To sum up, I personally like this paper. It is well written with sufficient evidence. Some issues, nevertheless, should be carefully addressed for a clear presentation. Hope these suggestions help.

Reviewer 2 Report

Dear Authors,

Thank you for the opportunity to review your manuscript. Below you will find a list of suggestions from a constructive point of view.

L108: I think height is expressed in cm, rather than in m

L109: At which level do this team compete? elite? sub-elite? national or international?

L113: Please provide IRB number.

L128: In order for this test to be replicated in the future, we would need the ball speed, landing point, arc, and frequency of the ball machine.

L132: Which force plate?

L132: What is anatomical posture?

L141: Why did you use this markerset? Does it have a name (for instance, Helen Hayes)?

L162: How did you measure the joint moment with the mocap system?

L194: Please provide more information about this software.

L202: The authors used parametric measures of central tendency and dispersion, meaning that the sample followed a normal distribution. Please acknowledge that in the manuscript.

L204: Are these angles measured in degrees? The t value of the t-test is not usually shown, but only p-value. 

L204: How many executions of these actions were considered in the mean+-SD values of each table?

L204: Also, for this table and the rest, effect sizes should be indicated to show the amount of practical significance of the difference between cross court and down the line.

L252: How joint moment is measured? Why so many decimals? It is hard to understand,

L269: The font size used in axes is too big. Also, phrase should read as phase.

L386: The entire discussion needs much more references to which compare your results. The number of references is too low for the statements you made.

Reviewer 3 Report

The purpose of this study was to compare the joint angle, angular velocity, and moment of the racket arm in backhand topspin strokes differences between the cross-court and down-the-line. This is a very great study. However, I recommended a few corrections prior acceptance for publication.

-Introduction: is too long. Please, shorter it.

-Methods: To add the number of local ethics committee. And The statistical analysis section is poor. What is sample size calculus? What is t value in the tables? 

-Results: There are too many figures.

-Discussion and Conclusion: Please to remove the first paragraph of conclusion.

Reviewer 4 Report

The main aim of the paper „Biomechanical comparison between down-the-line and cross-court topspin backhand in competitive table tennis“ is to compare the kinematic and kinetic differences of the racket arm when balls were hit cross court and down the line by topspin backhand. The study provides interesting suggestions for practice.

The study is interesting, meticulously processed, with interesting biomechanical analyses. I would like to applaud the efforts of the authors. There are just a few details that need to be explained.

Major comment:

Line 107: Please indicate how the study participants were selected.

Minor comments:

Table 5: In the first row, in the 3rd column the decimal point is missing: -00005388±0.00160. The last column is not aligned.

Round 2

Reviewer 2 Report

Dear Authors,

Thanks for all changes made to the manuscript.

Best regards

X